# Glucose Data Classification for Diabetic Patient Monitoring

**Amine Rghioui [1,\*], Jaime Lloret [2] , Lorena Parra [3], Sandra Sendra [2] and Abdelmajid Oumnad [1]**

[1]   Research Team in Smart Communications-ERSC–Research Centre E3S, EMI. Mohamed V University,
    Rabat 10000, Morocco; aoumnad@emi.ac.ma
[2]   Integrated Management Coastal Research Institute, Universitat Politecnica de Valencia,
    46370 Valencia, Spain; jlloret@dcom.upv.es (J.L.); ssendra@ugr.es (S.S.)
[3]   University of Granada, 18014 Granada, Spain; loparbo@doctor.upv.es
\*   Correspondence: rghioui.amine@gmail.com

**Abstract:** Living longer and healthier is the wish of all patients. Therefore, to design effective solutions for this objective, the concept of Big Data in the health field can be integrated. Our work proposes a patient monitoring system based on Internet of Things (IoT) and a diagnostic prediction tool for diabetic patients. This system provides real-time blood glucose readings and information on blood glucose levels. It monitors blood glucose levels at regular intervals. The proposed system aims to prevent high blood sugar and significant glucose fluctuations. The system provides a precise result. The collected and stored data will be classified by using several classification algorithms to predict glucose levels in diabetic patients. The main advantage of this system is that the blood glucose level is reported instantly; it can be lowered or increased.

**Keywords:** Internet of Things; Big Data; healthcare; machine learning; diabetes; blood glucose

---

## 1. Introduction

The presence of Internet of Things (IoT) technology is growing in several important technical areas like healthcare, smart houses, etc., and nowadays these applications are more and more accessible, available, and versatile, and allow the integration of the interconnected objects. The most important step in the field of intelligent health care is the quality of life of patients. For this reason, it is more important to build a new application in this field [1]. Daily mobile healthcare service is becoming more and more important. Chronic diseases influence the health of the people living with them, such as cardiovascular and diabetic diseases [2,3]. From patient data, analyses can be used to identify people who need "proactive care" or those who need a lifestyle change to avoid a deterioration in their health status. Patients do not have a problem giving away part of their privacy if this can help to save their lives or other people's lives. "If having more information about me enables me to live longer and be healthier", said Marc Rotenberg of the Electronic Privacy Information Center, "then I do not think I am going to object. If having more information about me means my insurance rates go up, or that I'm excluded from coverage or other things that might adversely affect me, then I may have a good reason to be concerned" [4].

Diabetes is a dangerous disease that can affect humans. Many deaths in the worlds are caused by diabetes, and most people are not yet aware of its danger. In addition, diabetes has occurred when an increase or a decrease in the level of glucose in the blood emerges. Therefore, it is necessary to keep it stable at a precise interval [5]. The growth of diabetic patients increased the use of continuous glucose monitoring devices (CGM), which are becoming the new method of continuous monitoring. They provide real-time information about the glucose level, which is updated every five minutes [6].

A glucose concentration value ≤70 mg/dL is defined as hypoglycemic and a glucose concentration value ≥180 mg/dL is defined as hyperglycemic. Periodic blood glycemic checks and consultation by medical experts is very important for patients. Then, to give an end to all these difficulties, a mobile blood glycemic device is required to send the glucose value to the medical experts. In other words, with the help of a tele-monitoring application, the problem can be solved. The tele-monitoring platform is based on SMS services via the GSM network that can be widely accessible.

In this paper, a smartphone-based wireless blood glucose monitoring system is developed. The measurement of blood glucose will be sent to the mobile phone through the wireless communication protocol (Bluetooth, Wi-Fi, Zigbee, RF). After that, the given information will be shared on a mobile or a web application. The purpose of this paper is to discuss the implementation of a digital blood glucose meter system integrated with the wireless network for monitoring a patient's glucose level in real-time. The patient can do the measurements by himself/herself at any place; the glucose level sensor will automatically send the data to the medical experts via SMS (Short Message Service). This application is firstly designed to help diabetic patients; secondly it is to provide periodic communication with the medical expert; and finally, it is intended for certain emergency cases and real-time applications. In addition to all these benefits, the medical expert could often monitor the patient's health remotely and provide early treatment if something happened to the patient. A healthcare data analytics system can help patients avoid several serious health problems, and can greatly influence the cost of patients for individual treatment [7].

The main contents of this paper are organized as follows; Section 2 resumes the description of current research on the application of blood glucose level measurement for diabetics. Section 3 describes the basic theory of the digital blood glucose meter, e-health module, and GSM module. Section 4 presents a description of the system implementation. Section 5 provides a brief description of the result discussed. Finally, a conclusion and future works are given in Section 6.

## 2. Related Work

This section presents some of the existing recent work related to e-health and monitoring glucose for diabetic patients.

Islam et al. [8] presented a survey in IoT-based health care technologies and reviewed the state-of-the-art network architectures/platforms, applications, and industrial trends in IoT-based health care solutions. Lloret et al. [9] presented an intelligent communication architecture for ambient assisted living (AAL). The process of artificial intelligence was used to gather the information from several types of communication and showed the main intelligent algorithms included in the AAL system; furthermore, a software application was developed. Finally, they used several real measurements to validate their proposal operation.

Monteiro et al. [10] presented in progress e-health architecture using IoT for data acquisition, fog for data pre-processing and short-term storage, and cloud for data processing, analyze and long-term storage. This work also describes the main challenges to provide an e-health application with high availability, high performance, and accessibility, at low deployment and maintenance cost. Xiao et al. [11] proposed real-time glucose monitoring for an electrochemical sensor based on a wireless system. A simple sensor tag with low power and low-cost solutions were contained in the implanted glucose monitoring application. The accurate results provided by the system and its use of real-time application were the advantages.

Wang and Lee [12] worked on development of a new blood glucose sensor. This sensor had the role of controlling and monitoring the level of glucose for diabetic patients. The real-time measurement of the blood glucose level was measured by a continuous glucose monitoring system (CGMS). This method used an alarm to indicate the level of the glycemic event as well as to improve the quality of life. Ahmed [13] designed a system for glucose concentration prediction of diabetic patients to avoid hypoglycemia or hyperglycemia of serious complications. They used GlucoSim software to analyze the information of the patients. It was generated by the continuous glucose monitoring system

(CGMS), which adopted a Kalman Filter (KF) to reduce noise. In [14] Siddiqui et al. presented a comprehensive survey on non-invasive/pain-free blood glucose monitoring methods from the five year period, 2012–2016. The devices manufactured or being manufactured for non-invasive monitoring were also compared in this paper. Giancarlo et al. [15] proposed a system architecture, Body Cloud, that integrates BSN services with a cloud computing infrastructure. This architecture supports the storage and management of sensor data streams and the processing (online and offline analysis) of the stored data using software services hosted in the cloud.

In our paper, we propose an IoT application based patient monitoring system and diagnostic prediction tool for diabetic patients. Most people with diabetes die because they do not receive appropriate medical care at the right time. The development in the field of sensors and the Internet of Things helps to constantly monitor the health parameters of patients. For those reasons, we use a simple sensor with low power and low-cost solutions for the glucose monitoring application. The patient's data is updated in the cloud every day. The collected data is used by the doctor to monitor the patient's blood glucose variation visually and protect the patient's health from a critical condition. With the help of sensors, the current health parameters of patients are collected and forwarded to the cloud for storage. Each health dataset is analyzed and verified to determine precisely if the current values are within the normal range or not. An SMS is sent to notify the doctor in the case of any change in the vital signs. The critical data is then sent directly to the prediction system. The system will consider this data as trial data and predict whether or not the patient has a high risk of stroke. The prediction is used with a Decision Tree algorithm (one of the complex algorithms in the fields of pattern recognition and data mining, which is not only related to databases, artificial intelligence, and other disciplines but has a great theoretical research value). This method can be improved to increase the accuracy value. We use also the naïve Bayes classifier called Bayesian Classification (a statistical classification method that can be used to predict the probability of membership of a class). The naïve model is developed by using the classifications of algorithms in machine learning. Different classification algorithms are analyzed for this model; the one that gives the best accuracy is considered for the proposed system. Ensemble algorithms gave better accuracy for the prediction.

The prediction system used in our application provides a good result for the first time and is used in a real-time application.

## 3. Proposal Description

This section presents a detailed description of the proposed blood glucose monitoring system. The aim of this project is to monitor the blood glucose level of the diabetic patient. The measured data is updated in the cloud every day. This collected data is used by the doctor or another medical expert to monitor the patient's blood glucose variation visually and send the response in real-time to protect the patient's health from a critical condition.

The designed blood glucose monitoring system consists of three distinct subsystems (Figure 1). The first is dedicated to capturing glucose level in the blood with the glucose sensor. This sensor captures the level of glucose in the blood, displays its exact value, and transmits it via the communication module (Bluetooth, Wi-Fi, 4G) to the Android mobile phone. This is necessary because current mobile phones do not include means of direct interface with the analog signals of the outside world. The second subsystem includes the storage and data processing service received by the smartphone. The third subsystem is data analysis by domain specialists and decision-making.

For a better diabetes management strategy, the glucose sensor will provide the diabetic patient and health professionals with the necessary information about diabetes. The monitoring of the blood glucose levels can help to reduce a person's risk with a range of diabetes-related complications. Structured self-monitoring involves checking the blood glucose levels at certain times of the day for a given period (i.e., two weeks), and then working with the diabetes healthcare team to determine how food, physical activity, and treatment have an impact on the blood glucose levels.

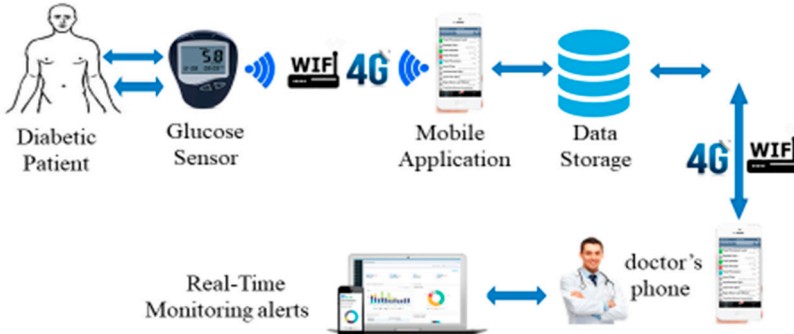

**Figure 1.** Proposed scenario.

## 4. Patient Monitoring Elements

### 4.1. E-Health Sensor Shield

The e-health sensor shield can be connected to an Arduino board for all electronic projects requiring the use of reliable biometric measurements. It provides information collected from nine different biometric sensors. By collecting an extensive set of biometric measures, one can then analyze this data, either as part of a medical diagnosis, or for monitoring in real-time, or for gathering data for later analysis. The data collected can be transmitted via the shield's various interfaces (Wi-Fi, GPRS, Bluetooth, and 3G, 802.15.4 or ZigBee). It is used for a communication interface between the sensors (Blood Pressure meter) and an Arduino board.

### 4.2. Digital Blood Glucose Meter

Each patient requires a different blood glucose testing approach for their diabetes, and some will need to test blood glucose more regularly than others. The measurement of blood glucose is a measure of the level of glycemia in the blood using an electronic device called a blood glucose meter. To do this, the patient pricks their fingertip with a small needle and places a drop of blood on a test strip. The strip is inserted into the blood glucose meter. The digital display shows the blood sugar level shortly afterward. There is a wide variety of blood glucose meters on the market. Some meters are designed for ease of use, others for ease of transport or connectivity, and still others incorporate more advanced technology such as USB software or multi-strip testing. Figure 2 shows the Blood Glucose Meter.

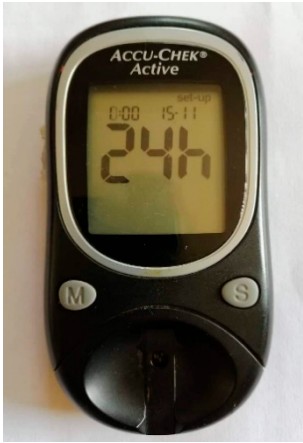

**Figure 2.** Digital blood glucose sensor.

*4.3. GSM Modem*

The GSM module is a compatible Arduino interface card equipped with a SIM card, which connects to the telephone network as a mobile phone that has its own phone number and operates anywhere in the world where there is a GSM cellular network. The GSM module enables the Arduino board connected to a mobile network to send and receive SMS either in the form of text or in the PDU format. The process of sending and receiving an SMS on a Wavecom modem uses an AT command. Figure 3 shows the GSM modem with External Antenna.

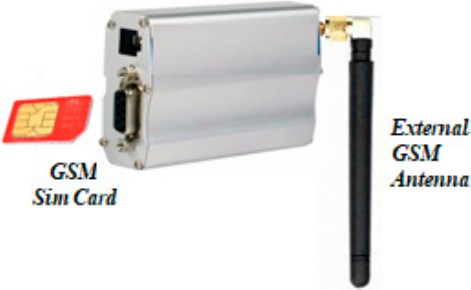

**Figure 3.** GSM modem.

*4.4. Arduino Card*

An Arduino board is an electronic card that can be programmed to analyze and produce electrical signal, to perform a variety of tasks such as home automation, driving a robot, embedded computing, etc. It is a platform based on a simple input/output interface. In addition, Arduino cards have serial communication interfaces, including a universal serial bus (USB) on some models, which are also used to load programs from personal computers. Figure 4 shows the Arduino Uno card.

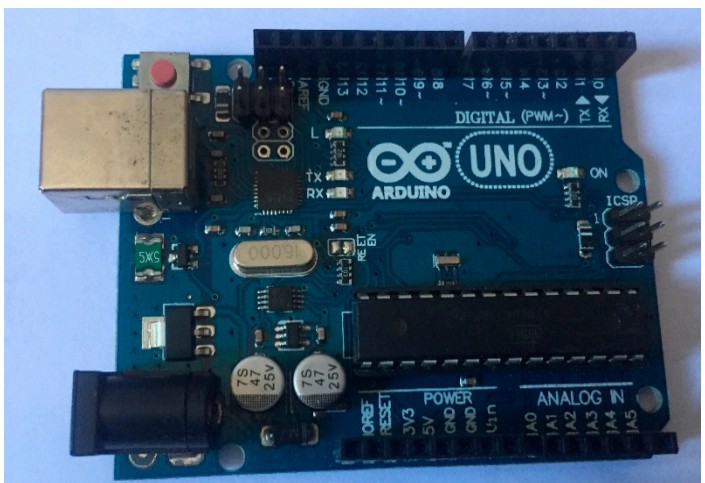

**Figure 4.** Arduino card.

## 5. Implementation

This section describes in detail the implementation of our proposed system including the scheme, system illustration, hardware, and software installation.

The flowchart of the system in Figure 5 shows the complete system from the measurement of glucose level in the patient until the final decision of the doctor.

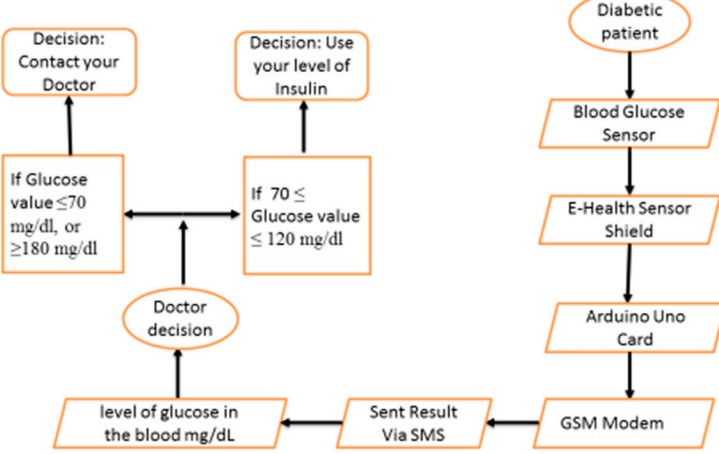

**Figure 5.** Flow chart of the system.

In this system, we describe the general scheme of the proposed system (Figure 6). The patient can perform a measurement of blood glucose using the blood glucose sensor. After the measurement, the results of the glucose level in the blood will be obtained for the patient. The measurement data is automatically transmitted by the blood glucose sensor that is connected in serial mode with the Arduino card. The data processed by the Arduino card's microcontroller is automatically transmitted to the GSM modem that has been integrated into the Arduino board to send the measurement results to a specialist doctor via SMS.

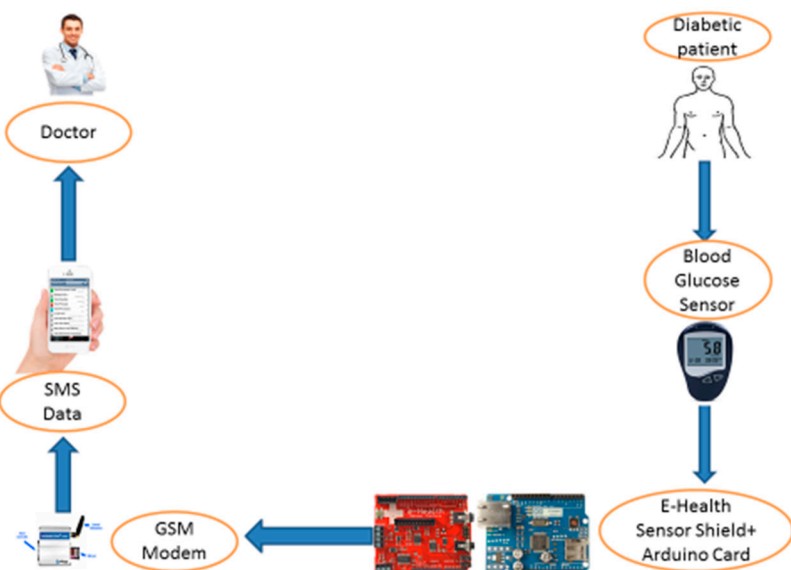

**Figure 6.** Illustration of blood glucose measurement.

### 5.1. Hardware Installation

The general scheme of the proposed system is achieved by connecting the blood glucose level sensor with the e-health shield and the Arduino board. The Arduino board is also connected to the GSM module with a TTL (Transistor-Transistor Logic) to RS232 converter. Data rates for serial communication between the microcontroller, the sensor, and the GSM module have a value of 115,200 bps. In addition, the type of data transmitted in the program must be also set to synchronize the transmitter and the receiver. Figure 7 shows the global hardware installation.

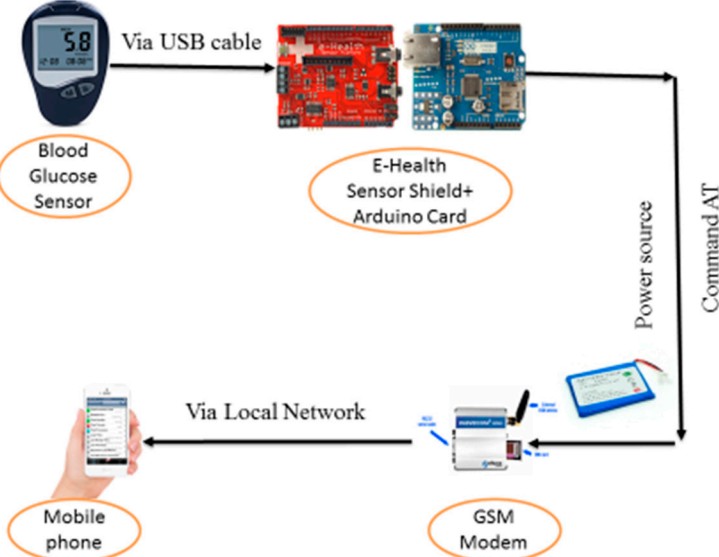

**Figure 7.** Hardware installation.

### 5.2. Software Installation

The measurement data from the sensors are sent via the serial data to the Arduino shield e-health. Finally, the serial data is processed by the microcontroller of the Arduino board and sent via the GSM modem. The processed data is then sent to the doctor via the SMS service. The data sent is displayed and processed by the doctor to make a decision about each patient. The Arduino program of our application is shown in Figure 8.

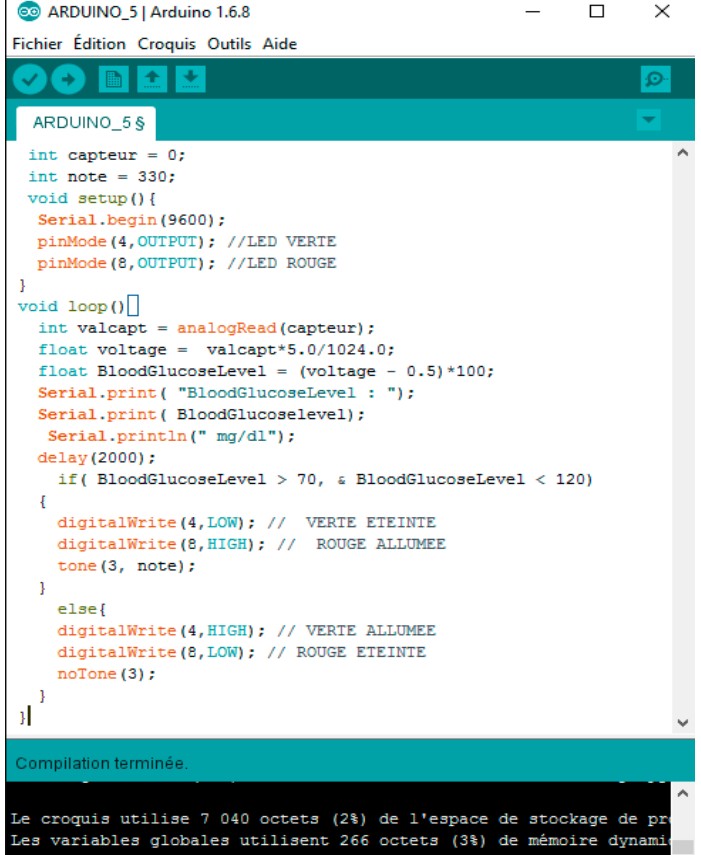

**Figure 8.** Coding for the reading of the sensor.

## 6. Classification Data in Healthcare

Classification is a type of supervised learning that consists of classifying data into a category. Classification involves predicting a certain outcome based on a given input. In order to predict the results, it needs to fetch the data already available. There are several classification models, namely, decision tree, ripper rule, neural networks, naïve Bayes, k-nearest neighbors, and support vector machine. In this section, we present the decision tree algorithm, naïve Bayes algorithms, random forest algorithms, and neural networks.

In [11], the authors used classification techniques to find out patterns from the diabetes data sets. They employed naïve Bayes and decision tree algorithms by using the Weka tool. For the analysis of diabetic data, many authors preferred the decision tree for the classification. In this work, we are using naïve Bayes, random tree, ZeroR, and J48 classification algorithms from the diabetes data set to test the most powerful to determine the patient's level of risk.

### 6.1. Decision Tree

A decision tree is a widely used classification technique. The decision tree is one of the classification techniques in which classification is done by the splitting criteria.

There are many related works in the literature about classification data with the decision tree algorithm. Kumari et al. [16] used a decision tree to predict cardiovascular disease, and they classified and diagnosed the patients. Al-Radaideh et al. [17] applied a decision tree method to predict the grades of students. Even though various classification algorithms were applied to the same, decision trees gave better results.

### 6.2. Naïve Bayes

The naïve Bayes classifier is a simple probabilistic classifier based on applying Bayes' theorem with strong (naïve) independence assumptions. The approach used in naïve Bayes classifier is very simple. The advantage of the naïve Bayes classifier is that with the help of a small amount of training data, it is possible to classify the given instances [18].

### 6.3. Random Forest

Random Forest is a machine-learning algorithm; it is a flexible and easy-to-use algorithm that produces, even without hyper-parameter settings, an optimal result most of the time. It is also one of the most used algorithms because of its simplicity and the fact that it can be used for both classification and regression tasks, because with random forest, one can also handle task regression using the random forest regressor. Random forest has almost the same parameters as a decision tree. In the field of health care, it is used to identify the correct combination of components in medicine and to analyze a patient's medical history to identify diseases. Finally, in e-commerce, random forest is used to determine whether a customer will really appreciate a product or not.

The data measured by the glucose sensor are processed by the Arduino card and sent directly to the patient's smartphone via Wi-Fi or 4G, which in turn sends these data daily to the cloud and to the smartphone of the doctor. The monitoring center has the right to access data from the cloud via local connection. Table 1 shows the value of glucose levels measured with the glucose sensor and processed by the Arduino Card into 15 days for one diabetic patient with an average of two times per days.

**Table 1.** Glucose measurement.

| Day | Morning | Evening |
|-----|---------|---------|
| 1 | 1.34 | 0.78 |
| 2 | 2.43 | 0.87 |
| 3 | 1.11 | 1.02 |
| 4 | 2.45 | 0.55 |
| 5 | 0.88 | 1.43 |
| 6 | 1.00 | 0.78 |
| 7 | 0.99 | 1.34 |
| 8 | 1.32 | 1.03 |
| 9 | 1.09 | 1.01 |
| 10 | 1.34 | 1.22 |
| 11 | 0.89 | 0.87 |
| 12 | 1.22 | 1.15 |
| 13 | 1.41 | 1.33 |
| 14 | 1.06 | 0.94 |
| 15 | 0.91 | 1.05 |

## 7. Result and Discussion

In this section, we show and discuss the test results of our proposed system; the plan for this research is to evaluate the performance of methods for classifying diabetes data. The data is evaluated using the J48, naïve Bayes, RandomTree, and ZeroR algorithms. We used a dataset that included 30 patients for a total of 30 days: 23 men and 7 women aged 40 to 60 years. The dataset used for the purpose of analysis had a total of 2700 instances, in which there were 2070 instance of men detected and 630 of women detected. The format of this dataset contains five columns designated by date, day, glucose level, and request. This research work deals mainly with the accuracy of classification algorithms with respect to execution time and error rate with WEKA software [19]. To classify diabetes data appropriately from the calculation dataset, error rates and accuracy are calculated using classifiers. The results of various measurements are given in Table 2.

**Table 2.** The accuracy level.

| Algorithms | Correctly Classified Instances | Incorrectly Classified Instances |
|-----------|-------------------------------|----------------------------------|
| Naïve Bayes | 91.8301% | 8.1699% |
| RandomTree | 99.6732% | 0.3268 |
| ZeroR | 69.6078% | 30.3922 |
| J48 | 99.3464 | 0.6536% |

This section of the paper focuses on predicting the possible number of diabetic patients from the dataset using data classification techniques and determining which model gives the highest percentage of correct predictions for the diagnoses. These techniques are required for prediction of diabetes in patients.

This research has the major goal of predicting the number of diabetics and comparing the best available methods of prediction. Therefore, we can determine what attribute is the best predictor for the correct prediction of the diagnoses, and we can help experts to predict possible diabetic attacks from the patient datasets.

From the test results, as shown in Table 1, it can be stated that the measurement data were successfully sent. Among 20 times of testing, the range of delay between transmitting and receiving SMS was from 22 to 91 s, and the average was 32.17 s. Figure 9 shows a graph of the measurement of the level of glucose in the blood.

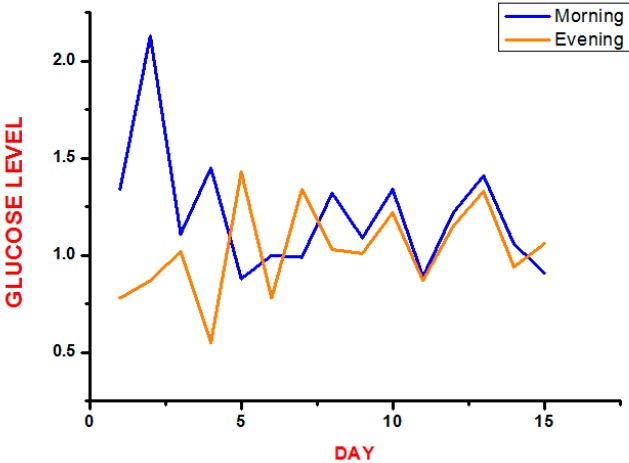

**Figure 9.** Measurement of the level of glucose in the blood.

The test of the dataset was analyzed by the use of four algorithms classification, which were the naïve Bayes, J48, ZeroR, and RandomTree (using training set). In addition, a comparison of the accuracy of all classifiers was done and finally, it was investigated that the RandomTree technique performed better with an accuracy of 99.6732%. The accuracy level of all the algorithms is given in Table 2.

As seen in Table 2, the comparison between the algorithms like ZeroR, RandomTree, J48, and naïve Bayes for the correctly classified instances was higher than the incorrectly classified instances in terms of accuracy. Figure 10 shows a bar graph that represents correctly and incorrectly ranked instances of the algorithms used.

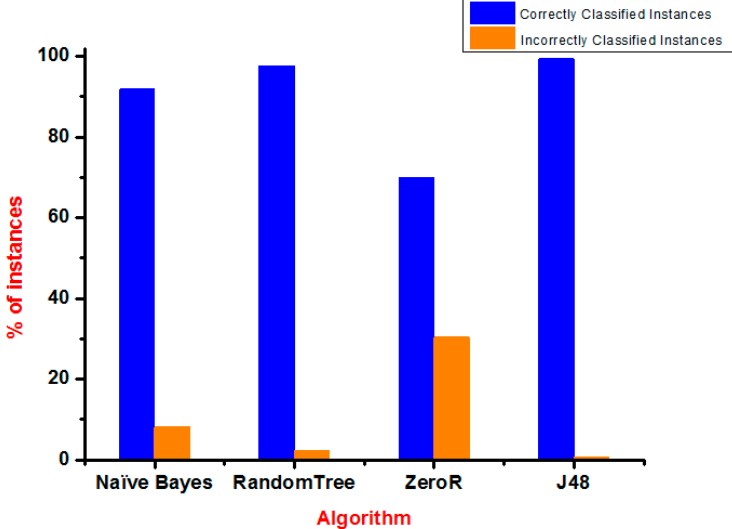

**Figure 10.** The graph of correctly and incorrectly classified instances of algorithms.

Figure 11 shows the time graph of various classification algorithms. The longest time was taken by ZeroR consuming a time of 0.05 s and the shortest time was taken by RandomTree consuming 0.01 s only.

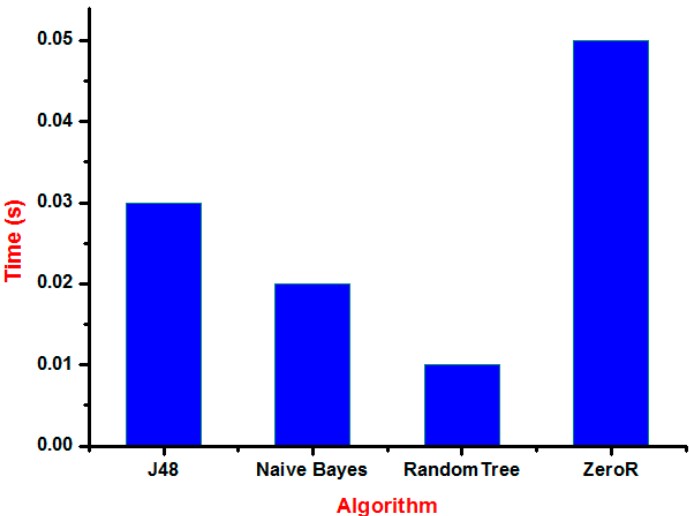

**Figure 11.** The graph for training time results for the top four classifiers.

Figure 12 represents Kappa statistic, MAE (mean absolute error), and RMSE (root mean squared error) values for the chosen classifiers. The naïve Bayes algorithm showed better results for MAE and RMSE, and the J48 algorithm showed a better value for the Kappa statistic.

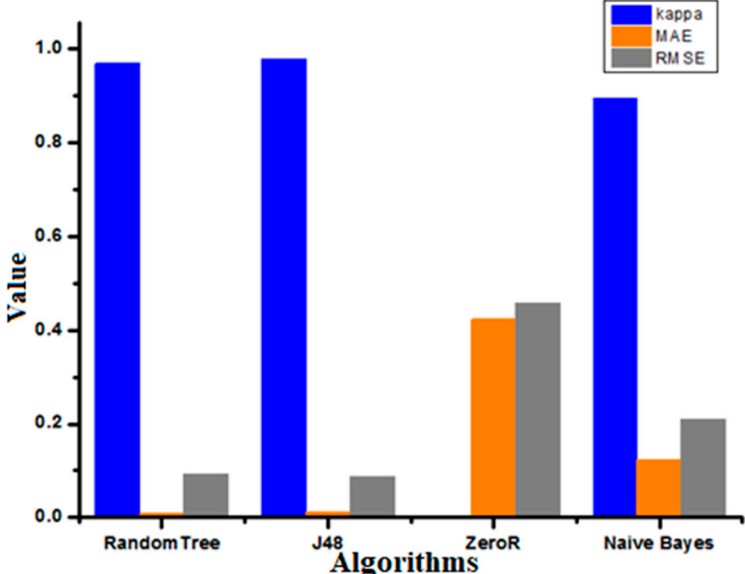

**Figure 12.** KAPPA statistics and error rate values.

Figure 13 shows the decision tree built by the classifier, which includes the root nodes, child nodes, and leaf nodes with their possible/predicted values.

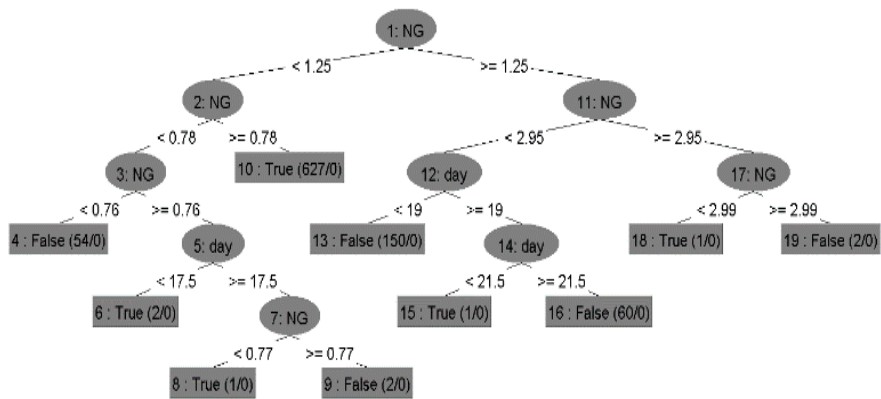

**Figure 13.** Visualized tree with random tree.

Figure 14 represents TP Rate, FP Rate, Precisions, Recall, and F-Measured for the naïve Bayes algorithm, Random Tree, ZeroR and J48.

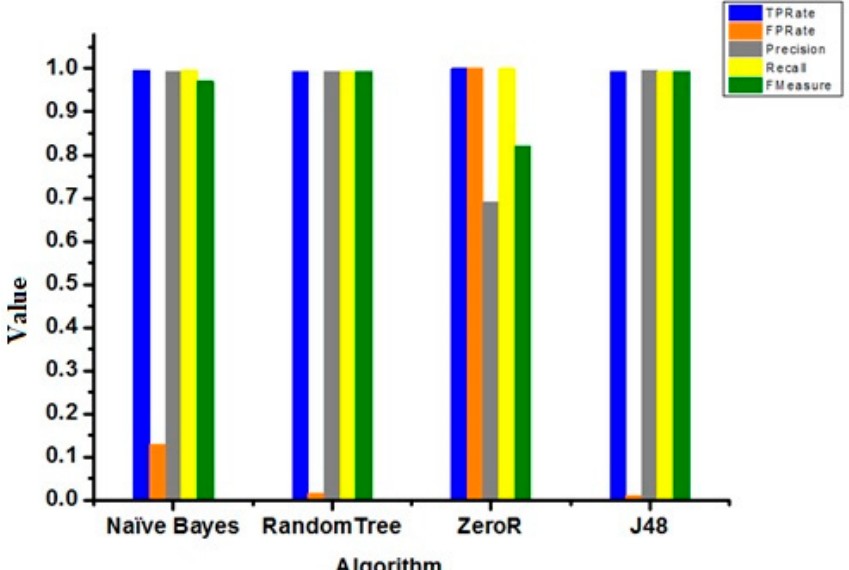

**Figure 14.** Graphical representation of false positive (FP), true positive (TP), precision, recall, and F-measure of different algorithms.

In the above section, all the rules were interpreted. These rules were visualized in the tree form. Tree visualization is the simplest method to understand the conditions and their results.

The performance of the proposed classification approach was evaluated using different standard measures based on true positive (TP), false positive (FP), precision, recall, and F-measure [20].

The accuracy of algorithms (shown in Table 3 and Figure 13) was measured with the help of parameters such as false positive rate, true positive rate, recall, precision, and F-measure. These parameters are defined as follows:

- True positive (TP): represents the instances that have been correctly classified with respect to a given class.
- False positive (FP): represents the instances that have been incorrectly classified with respect to a given class.
- Precision: the proportion of the correctly classified instances to a particular class divided by overall instances classified with respect to that class.

$$Precision = TP/(TP + FP) \qquad (1)$$

- Recall: the proportion of the incorrectly classified instances that have been classified by a class divided by the total instances present in the class.

$$\text{Recall} = TP/(TP + FN) \tag{2}$$

- F-Measure: it is calculated by combining the measure of recall and precision according to Equation (3):

$$\text{F-Measure} = (2 * \text{Precision} * \text{Recall})/(\text{Precision} + \text{Recall}), \tag{3}$$

**Table 3.** Values of TP, FP, precision, recall, and F-measure for algorithms.

| Algorithm | TP Rate | FP Rate | Precision | Recall | F-Measure |
|---|---|---|---|---|---|
| Naïve Bayes | 0.996 | 0.129 | 0.946 | 0.996 | 0.970 |
| RandomTree | 0.994 | 0.016 | 0.993 | 0.994 | 0.994 |
| ZeroR | 1 | 1 | 0.69 | 1 | 0.82 |
| J48 | 0.993 | 0.010 | 0.996 | 0.993 | 0.994 |

Figure 13 represents TP rate, FP rate, precision, recall, and F-measure for the chosen classifiers. The TP rate, precision, recall, F-measure values for J48 were better than the other classifiers, but the FP Rate for ZeroR was better than the others.

According to the precedent equations, we can obtain the sensitivity, specificity, and accuracy of different algorithms.

$$\text{Specificity} = TN/(TN + FP) \tag{4}$$

$$\text{Sensitivity} = TN/(TP + FP) \tag{5}$$

$$\text{Accuracy} = (TP + TN)/(TP + TN + FN + FP) \tag{6}$$

Table 3 shows the value of TP Rate, FP Rate, Precision, Recall, F-Measure, and Table 4 shows Sensitivity, specificity, and Accuracy for the Naïve Bayes algorithm, Random Tree, ZeroR and J48.

**Table 4.** Comparison of sensitivity, accuracy, and specificity.

| Algorithm | Sensitivity | Specificity | Accuracy |
|---|---|---|---|
| Naïve Bayes | 0.36 | 0.87 | 0.95 |
| RandomTree | 0.42 | 0.99 | 0.99 |
| ZeroR | 0 | 0 | 0.69 |
| J48 | 0.43 | 0.99 | 0.99 |

Figure 15 represents the sensitivity, specificity, and accuracy of different algorithms. The sensitivity, specificity, and accuracy for J48 were better than for other classifiers.

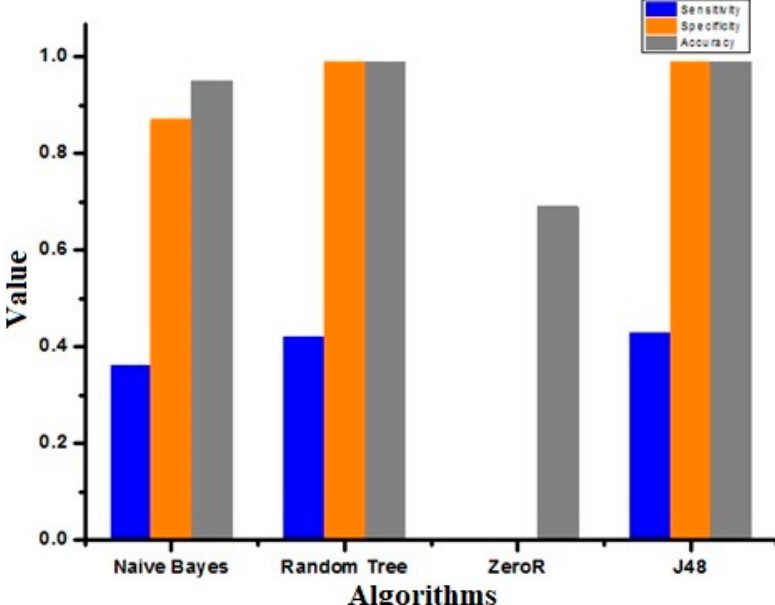

**Figure 15.** Graphical representation of sensitivity, specificity, and accuracy of different algorithms.

## 8. Conclusions

As given in the paper, the real-time monitoring system for the diabetes condition is able to perform with the function of the Internet of Things (IoT). The effectiveness of the personal diabetes monitoring via a mobile application database has been demonstrated. In addition to that, an implementation of the blood glucose measurement system for tele-monitoring is presented. Additionally, after several tests, the evaluation result showed that the system could work appropriately with a delay of transmission less than 32 s. The measured data was sent and received with a valid value by SMS. Finally, the proposed prediction system was evaluated by several simulation experiments and the results demonstrated that the proposed model improved the accuracy of prediction. This model allows a preventive intervention for a diabetic patient and controls the glucose.

The future work aims to send the blood glucose reading over the cloud so that it can be accessed from any location across the globe. Additionally, a new concept of early detection is planned in order to detect the symptoms of other diseases that need prompt intervention. In addition, the proposed system of monitoring will be enhanced with the inclusion of a content management framework to obtain additional information about physical activity, and communication with electronic health record (EHR) from the hospital information system to get relevant information such as illness, treatments, and drugs. Moreover, we will study the performance of our system and include algorithms to predict the connectivity of the devices in the wireless network [21].

**Author Contributions:** Methodology and supervision, J.L. and A.O.; formal analysis and investigation, A.R., L.P. and S.S.; writing—original draft preparation, A.R.; writing—review and editing, A.R. and J.L.

**Funding:** This work has been partially supported by the "Ministerio de Economía y Competitividad" in the "Programa Estatal de Fomento de la Investigación Científica y Técnica de Excelencia, Subprograma Estatal de Generación de Conocimiento" within the project under Grant TIN2017-84802-C2-1-P.

**Conflicts of Interest:** The authors declare no conflict of interest.

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
