# Peer review of "Glucose Data Classification for Diabetic Patient Monitoring"

_applsci, doi:10.3390/app9204459_

Round 1
Reviewer 1 Report
The paper describes a system aiming at real-time monitoring patients' glucose levels and classifying the patients on the basis of the risk of strokes.
The Introduction provides sufficient information to position the work in the context of telemedicine and e-healt.
The Related Works section briefly reviews some other research work concerning glucose monitoring. It is sufficient, by authors may review other papers covering e-healt and IoT topics.
Section 3 furnishes an overview of the systems main components, while Section 4 lists and briefly describe the exploited hardware. Section 5 provides information about developed software and system deployment.
Section 6 summarizes the set of well-known classification algorithms exploited for predicting patients' risk of strokes.
Section 7 presents a set of results for validating the effectiveness of the system. Authors show measures about the success rate and transmitting time of the communication between the smartphone and the server (through SMS). The authors should discuss why the proposed measures are interesting for validating the performance of the system.
Then the paper shows a series of results and comparison of the previously listed classification techniques applied to the data gathered by the system. Here a more detailed description of the dataset structure should be given.
The paper has merits but requires a deep language improvement. I suggest the authors to check the paper with a native English speaker.
Author Response
Thank you very much for your time providing these useful comments that have helped us to improve our paper.
Please, find below our reply explaining our improvements in the paper.
Point 1: The Related Works section briefly reviews some other research work concerning glucose monitoring. It is sufficient, by authors may review other papers covering e-health and IoT topics.

Response 1: We have improved the Related Work section adding more papers covering e-health and IoT topics.
Point 2: Section 7 presents a set of results for validating the effectiveness of the system. Authors show measures about the success rate and transmitting time of the communication between the smartphone and the server (through SMS). The authors should discuss why the proposed measures are interesting for validating the performance of the system.
Response 2: a paragraph has been added explaining the interest of the proposed measures to validate the system performance.
Point 3: Then the paper shows a series of results and comparison of the previously listed classification techniques applied to the data gathered by the system. Here a more detailed description of the dataset structure should be given.
Response 3: This study enrolled 30 diabetes 23 men and 7 women aged 40 to 60 years, with 3 measurements per day, for 30 days.
Point 4: The paper has merits but requires a deep language improvement. I suggest the authors to check the paper with a native English speaker.
Response 4: We have improved language with an English speaker.
Reviewer 2 Report
This paper has proposed a system which can provide in time data of blood glucose levels. This could be very helpful to prevent and monitor diabetics health conditions.
However, the whole paper needs to be better organized. For example, the introduction is too long. Instead, the paper should focus more on the system by emphasizing the data storage and transfer.
Author Response
Thank you very much for your time providing these useful comments that have helped us to improve our paper.
Please, find below our reply explaining our improvements in the paper.
Point 1: However, the whole paper needs to be better organized. For example, the introduction is too long. Instead, the paper should focus more on the system by emphasizing the data storage and transfer.

Response 1: We have improved the Introduction, and we have added a paragraph that explains the Data storage and transfer.